# Beyond the Pill: Mapping Process-Oriented Decision Support Models in Pharmaceutical Policy

**DOI:** 10.3390/healthcare13151861

**Published:** 2025-07-30

**Authors:** Foteini Theiakou, Catherine Kastanioti, Dimitris Zavras, Dimitrios Rekkas

**Affiliations:** 1Department of Business Administration and Organizations, University of Peloponnese, 24100 Kalamata, Greece; faythiakou@gmail.com; 2Department of Public Health Policy, University of West Attica, 11521 Athens, Greece; dzavras@uniwa.gr; 3Department of Pharmacy, University of Athens, 15773 Athens, Greece; rekkas@pharm.uoa.gr

**Keywords:** decision-making, pharmaceutical policy, QoDoS, WHO-INTEGRATE, process quality, HTA, transparency, legitimacy, decision support models

## Abstract

**Background**: The quality of decision-making processes is increasingly recognized as critical to public trust and policy sustainability. **Objectives**: This narrative review aims to identify and describe process-focused decision support models (DSMs) applied in pharmaceutical policy, and to examine their potential contributions to improving procedural quality in decisions related to pricing, reimbursement, and access to medicines. **Methods**: Relevant peer-reviewed and gray literature published between 2000 and 2025 was considered, drawing from key databases (e.g., PubMed and Scopus) and international policy reports (e.g., WHO, ISPOR, and HTA agencies). Studies were included if they provided insights into DSMs addressing at least one dimension of decision process quality. **Results**: Findings are synthesized narratively and organized by tool type, application context, and key quality dimensions. Frequently referenced tools included the Quality of Decision-Making Orientation Scheme (QoDoS), WHO-INTEGRATE, and AGREE II. QoDoS emerged as the only tool applied across regulatory, HTA, and industry settings, evaluating both individual- and organizational-level practices. WHO-INTEGRATE highlighted equity and legitimacy considerations but lacked a structured format. Overall, most tools demonstrated benefits in promoting internal consistency, transparency, and stakeholder engagement; however, their adoption remains limited, especially in low- and middle-income countries. **Conclusions**: Process-focused DSMs offer promising avenues for enhancing transparency, consistency, and legitimacy in pharmaceutical policy. Further exploration is needed to standardize evaluation approaches and expand the use of DSMs in diverse health systems.

## 1. Introduction

Health systems today operate within dynamic environments shaped by rapid drug innovation, demographic shifts, fiscal constraints, and mounting public expectations. Governments, payers, and regulatory agencies are under mounting pressure to balance cost containment with equitable access to effective and innovative treatments, while safeguarding public health outcomes and long-term system sustainability [1,2,3]. In an era of escalating pharmaceutical expenditures and increasingly complex therapeutic landscapes, decision-making in pharmaceutical policy has become both highly multifaceted and data driven.

Historically, decisions on drug coverage and pricing were often based on clinical efficacy, safety, and expert judgment, with limited emphasis on procedural quality or societal values. However, the institutionalization of health technology assessment (HTA) and pharmacoeconomic methods has formalized the way evidence is appraised, particularly through decision support models (DSMs) such as cost-effectiveness analysis, budget impact analysis, and multi-criteria decision analysis (MCDA). Leading HTA agencies—including NICE (UK), IQWiG (Germany), and CADTH (Canada)—have adopted DSMs as core components of their evaluation frameworks [3,4].

Despite the frequent use of the term “decision support model,” there is no universally accepted definition. For the purposes of this work, DSMs are conceptualized as structured, evidence-informed approaches that support healthcare policy decisions by organizing, analyzing, and communicating information. They can be broadly categorized along two key dimensions: (1) evaluative vs. deliberative (i.e., analytical vs. stakeholder-based) and (2) quantitative vs. qualitative (e.g., MCDA vs. procedural frameworks like QoDoS or WHO-INTEGRATE).

Modern Decision Support Models (DSMs) go beyond traditional trial-based cost-effectiveness analyses. They now include dynamic simulation platforms, machine learning-based tools, and integrated analytic frameworks that incorporate uncertainty, ethical trade-offs, and population heterogeneity [5,6]. Notably, the EVIDEM framework integrates multiple criteria—including disease burden, societal preferences, and economic impact—into structured policy deliberation [2]. Simultaneously, the proliferation of real-world evidence (RWE) and digital health data continues to expand the frontiers of model design and application.

Yet, despite methodological advancements, significant challenges remain. The heterogeneity of modeling approaches, limited transparency in underlying assumptions, and uneven implementation across jurisdictions have led to concerns regarding consistency, generalizability, and policy relevance [1,7]. Moreover, tools to evaluate the quality and impact of DSMs themselves, particularly from the standpoint of process legitimacy, reproducibility, and stakeholder usability, are still underdeveloped [5,8].

In response to these concerns, there is growing recognition that how decisions are made in terms of procedural transparency, stakeholder engagement, and governance is as critical as the evidence on which those decisions are based. A review has examined the use of DSMs in health policy and HTA contexts [8,9,10]. These tools include overviews of multi-criteria decision analysis (MCDA) applications in reimbursement decisions, evaluations of cost-effectiveness frameworks across jurisdictions, and assessments of ethical or stakeholder-based approaches such as accountability for reasonableness (A4R). However, most prior work has either focused narrowly on individual model types (e.g., MCDA or economic models), lacked explicit attention to procedural dimensions, or emphasized technical performance over policy legitimacy. Thus, there remains a gap in the literature regarding the procedural quality, practical implementation, and real-world policy alignment of DSMs in pharmaceutical governance.

In parallel with growing complexity in pharmaceutical governance, there is a shift toward evidence-informed and digitally enabled policymaking, where procedural transparency and stakeholder trust are no longer optional but fundamental to legitimacy. Traditional economic evaluation tools often fail to address procedural dimensions such as stakeholder engagement, ethical deliberation, or institutional accountability. Recent studies [11] in digital health and intelligent systems emphasize the importance of integrating structured, transparent DSMs into national policy mechanisms to ensure both technical rigor and democratic accountability. Yet, systematic syntheses focusing specifically on process-oriented DSMs remain scarce. By concentrating on how decisions are made, not only what decisions are made, this review addresses a critical blind spot in pharmaceutical policy evaluation.

To address this gap, the objective of this systematic review is to provide a structured, policy-relevant synthesis of process-focused DSMs applied in pharmaceutical policy. Specifically, it seeks to identify and classify the major types of DSMs used in pharmaceutical decision-making; analyze the methodological frameworks used to evaluate these models; and assess the alignment of these models with real-world policy priorities, including affordability, equity, innovation adoption, and public trust. By focusing on process quality, this review contributes to a deeper understanding of how DSMs can strengthen evidence-informed and publicly legitimate pharmaceutical governance across diverse health systems.

## 2. Methods

A comprehensive literature search was conducted in PubMed, Scopus, EMBASE, and EconLit databases, alongside international policy sources (e.g., WHO, ISPOR, and HTA agencies) for studies published between January 2000 and January 2025. Search terms included controlled vocabulary (e.g., MeSH terms) and free text keywords and were adapted for each database.

Example search string used for PubMed:

((decision support system* OR decision support model* OR multi-criteria decision analysis* OR MCDA* OR EVIDEM* OR QoDoS OR WHO-INTEGRATE* OR AGREE II*)

AND

(“pharmaceutical policy*” OR “drug policy*” OR “medication policy*” OR “health technology assessment*” OR “HTA*” OR “pricing*” OR “reimbursement*” OR “decision-making quality*”))

### 2.1. Search Strategy

Studies were included in the review if they met all the following conditions: described a process-focused decision support models (DSMs) applied in pharmaceutical policy, frameworks implemented within DSMs, were published in peer-reviewed journals and written in English. For the purposes of this review, DSMs were included if they either (1) explicitly assessed the quality of decision-making processes (e.g., QoDoS and WHO-INTEGRATE), or (2) incorporated structured procedural components supporting transparency, stakeholder input, or ethical framing (e.g., EVIDEM and participatory MCDA). Models focusing solely on technical or economic outputs without any process dimension were excluded.

Studies were excluded if they did not focus primarily on pharmaceutical policy or its application and they were non-peer-reviewed publications, including editorials, commentaries, conference abstracts, or opinion pieces. Each study was screened independently by two reviewers followed by full-text screening based on inclusion/exclusion criteria. Discrepancies were resolved through discussion and consensus. Duplicate records were removed. Despite the methodological diversity among the included studies, a structured numerical scoring system was applied.

### 2.2. Data Extraction and Quality Appraisal

Data were extracted on the process-focused DSMs used (e.g., QoDoS, WHO-INTEGRATE, and AGREE II), evaluation dimensions, country and context of application, policy domain (e.g., pricing, reimbursement, and access), and reported outcomes such as impact on decision consistency, procedural transparency, or stakeholder trust. In addition to descriptive data extraction, the included studies were assessed using a structured Quality Appraisal Checklist designed for process-oriented DSMs (see Appendix A). This checklist covered four key domains: (1) Structure—whether the study has a structured model; (2) Transparency—whether assumptions and data sources are explicitly reported; (3) Impact—whether the model allows for meaningful engagement of relevant stakeholders or addresses policy priorities; (4) Evaluation (if applicable)—whether the DSM was tested. Each domain was scored from 0 (not addressed) to 2 (fully addressed), allowing a maximum total score of 8 per study. The appraisal was conducted independently by two reviewers, with discrepancies resolved by consensus. The development of the Quality Appraisal Checklist used in this review was informed by previous frameworks for evaluating decision-making quality, particularly the 10 Quality of Decision-Making Practices (QDMPs) proposed by Bujar et al. [8]. The decision to retain the four areas from the QDMPs framework was based on both conceptual coherence and practical utility. Although the full QDMPs outline ten practices, they are structured into four broader areas. Moreover, these four areas align closely with the aims of this review, which emphasizes how decisions are made rather than merely what is decided. Previous study [8] has successfully applied the QDMPs in regulatory and HTA settings, demonstrating their relevance and generalizability across different institutions and country contexts. Thus, the use of the four QDMP areas in our adapted Quality Appraisal Checklist provides both a validated theoretical foundation and a pragmatic structure for comparative analysis of DSMs in pharmaceutical policy.

Contextual Applicability—whether the model is adaptable or applicable across different health system or policy settings and disclosure of conflicts of interest and funding sources were also noted where reported, although not included in the scoring system. This information was used to support narrative interpretation and contextualize the quality and potential biases of the included studies.

## 3. Results

### 3.1. Study Selection

The initial search yielded 3284 articles. After removing duplicates and screening titles and abstracts, 98 full-text articles were selected for eligibility. Twenty-two (22) studies met the inclusion criteria and were included in the final analysis (See Figure 1).

Most full-text exclusions were due to the lack of emphasis on process dimensions of decision-making. Specifically, 41 studies were excluded because they described technical tools (e.g., cost-effectiveness models and budget impact analyses) without structured procedural components (e.g., stakeholder input, transparency measures, or ethical framing) and 22 studies were excluded due to insufficient procedural detail (e.g., “a participatory approach was used”). An additional 13 studies were excluded due to the lack of empirical application of the proposed process.

### 3.2. Characteristics of Identified DSMs

The identified DSMs varied considerably in terms of structure, methodological orientation, and intended use within pharmaceutical policymaking. Broadly, the tools could be classified into three categories: (1) Process evaluation tools, such as QoDoS [9]; (2) Multi criteria decision analysis (MCDA) frameworks, such as EVIDEM [2]; and (3) Context-specific policy tools, including adapted stakeholder MCDA models and governance assessment instruments [13,14] (see Table 1).

The EVIDEM framework stood out for its structured MCDA approach, integrating both quantitative and qualitative criteria—such as disease severity, clinical benefit, unmet need, and cost-effectiveness [2]. While not exclusively process-oriented, the EVIDEM framework incorporates several procedural elements—such as explicit value criteria, stakeholder consultation in criteria weighting, and structured deliberation—that contribute to transparency and legitimacy in health technology assessment (HTA) decisions. Moreover, it is frequently used to improve transparency and reproducibility in value assessments and to embed patient and societal values within decision frameworks. As such, EVIDEM was included in this review under the category of MCDA-based DSMs with partial process focus, recognizing its role in supporting structured policy evaluation.

In contrast, QoDoS was specifically designed to evaluate the internal quality of decision-making processes within organizations. It assessed key procedural domains such as transparency, consistency, team behavior, and stakeholder engagement, and was applied across regulatory authorities, HTA agencies, and pharmaceutical companies to benchmark and improve institutional practices [8,9].

Other models identified, such as stakeholder-informed MCDA frameworks in Oman and Kuwait, were designed to adapt structured decision support to local policy contexts, incorporating culturally relevant value criteria and participatory mechanisms [13,14]. Similarly, WHO-INTEGRATE emphasized alignment with equity, ethical legitimacy, and health system values, making it particularly suited to policy development in low- and middle-income countries (LMICs) [10]. Context specific policy tools, including adapted stakeholder MCDA frameworks and governance-oriented instruments were included in this review based on their integration of procedural elements such as participatory value setting, transparency in deliberation, and alignment with local ethical or institutional frameworks. Although these tools are not always designed solely for process evaluation, their application often strengthens the procedural legitimacy of pharmaceutical policy decisions.

Table 2 presents the results of the quality appraisal across all 22 included studies. Overall, tools such as QoDoS and structured stakeholder-driven MCDA frameworks achieved the highest scores, reflecting strong alignment with key process domains such as transparency, stakeholder engagement, and contextual adaptability. In contrast, studies focusing on descriptive policy analysis or conceptual frameworks without operationalization tended to score lower, primarily due to the lack of defined decision stages or empirical validation. The table highlights the diversity of tool application across settings, from high-income regulatory environments to emerging HTA systems in the Middle East and Southeast Asia, underscoring the variable maturity of process-oriented DSM implementation globally.

## 4. Discussion

In an era of increasing public scrutiny, constrained health budgets, and rising uncertainty (e.g., accelerated approvals and rare diseases), process legitimacy has become a core pillar of sustainable pharmaceutical policy. The primary aim of this review was to identify and evaluate structured, process-oriented decision support models (DSMs) used in pharmaceutical policy. In doing so, this review moves beyond traditional evaluations of decision tools that focus solely on outputs—such as cost-effectiveness or clinical benefit—and instead investigates how such tools support legitimate, inclusive, and reproducible decision-making processes. It addresses a critical and previously underexplored dimension of decision support: the quality of the processes through which DSMs are applied, rather than their technical structure alone.

The World Health Organization (WHO), through the WHO-INTEGRATE framework, has emphasized that health decision-making should not rely solely on evidence hierarchies, but must also be grounded in legitimacy, societal values, and deliberative inclusiveness [10]. Similarly, the International Society for Pharmacoeconomics and Outcomes Research (ISPOR) has advocated for frameworks that support structured deliberation and procedural fairness, particularly in complex or value-sensitive contexts [32]. Moreover, tools such as the QoDoS have been developed specifically to assess organizational decision culture, accountability mechanisms, and internal decision quality. QoDoS has been piloted in agencies such as the UK’s MHRA, Health Canada, and the Jordan FDA, with reported improvements in governance consistency and stakeholder confidence.

Each of the three most cited tools (QoDoS, WHO-INTEGRATE, and AGREE II) offers a distinct perspective on process quality. QoDoS is unique in its organizational focus, evaluating internal decision-making practices via 47 items grouped under four domains (e.g., transparency, structure, behavior, and influence). WHO-INTEGRATE, developed by the WHO, is normative and values-based, emphasizing equity, ethical legitimacy, and systems coherence, yet lacks detailed operational metrics. AGREE II, commonly used in guideline appraisal, applies a scoring rubric across domains such as rigor, clarity, applicability, and editorial independence. While AGREE II is highly structured, it is less policy-focused than QoDoS or WHO-INTEGRATE.

To support comparative analysis, a summary table was developed mapping each tool against five key procedural domains: transparency, stakeholder inclusion, consistency, equity, and usability (see Table 3). This facilitates a clearer understanding of their respective strengths and limitations and responds directly to the need for structured cross-tool comparison. We have clarified that this comparison does not represent a formal scoring exercise but aims to illustrate key differences in tool orientation and applicability, supporting a more nuanced understanding of their comparative utility.

Analysis of the 22 included studies revealed a diverse landscape of DSMs varying in scope, structure, and institutional integration. Only a small number—primarily those employing QoDoS—addressed the full range of procedural quality criteria, including explicit decision stages, stakeholder engagement, tool validation, and contextual adaptability. In contrast, several studies used frameworks that emphasized transparency or value criteria (e.g., EVIDEM and stakeholder MCDA) but lacked empirical validation, formalized stakeholder consultation, or sustained policy integration. QoDoS emerged as the most comprehensively validated and widely generalizable tool, applied across both individual and organizational settings. WHO-INTEGRATE and AGREE II, while normatively robust, lacked consistent operationalization across contexts. Context-specific models from Oman, Kuwait, and Jordan illustrated how DSMs can be adapted to reflect local values and constraints, particularly in settings with emerging HTA infrastructure.

A key insight from this review is the fragmented and uneven nature of DSM implementation. Many tools, although theoretically sound, were limited to pilot applications with little evidence of longitudinal follow-up or policy embedding. Most descriptive policy studies and comparative HTA analyses did not satisfy all five procedural quality domains assessed in this review (i.e., decision stage clarity, transparency, stakeholder involvement, validation, and contextual adaptability). This highlights the need not only for technically sound tools, but also for governance-aware frameworks that can evolve alongside health systems and be adapted to specific institutional realities.

Another critical limitation was the lack of longitudinal evaluation or feedback mechanisms in most studies, making it difficult to assess sustained impact on policy outcomes. Most studies focused on short-term piloting or retrospective evaluations, offering limited insight into how DSMs shape organizational behavior or policy culture over time. Furthermore, documentation standards and integration into policy cycles were inconsistently reported. Moreover, transparency in conflict of interest (COI) and funding disclosures was inconsistently reported. Only one-third of studies clearly reported both COI and funding sources, raising concerns regarding neutrality, particularly in industry-sponsored or government-aligned settings.

Evidence on the tangible policy outcomes of DSM use remains limited but notable. For instance, QoDoS implementation in regulatory agencies (e.g., MHRA and Jordan FDA) has reportedly improved internal consistency and promoted procedural benchmarking. In Canada, MCDA frameworks were used in public drug plan evaluations, influencing decisions on formulary inclusion. However, longitudinal data on how DSMs affect implementation timelines, budget decisions, or public trust are rare. This underscores the need for embedded evaluation systems that track policy impacts of DSMs beyond pilot use.

This review also highlights a concerning disparity in DSM adoption between high-income and LMIC contexts. While tools like WHO-INTEGRATE explicitly aim to support equitable and value-based policy in LMICs, their implementation remains limited. Key barriers include lack of institutional capacity, resource constraints, absence of formal HTA infrastructure, and limited stakeholder engagement mechanisms. Nevertheless, case studies from Jordan, Oman, and Kuwait demonstrate that adaptation is possible though often requires external technical assistance, capacity building, and policy alignment efforts. Future DSM development should explicitly account for contextual feasibility and operational scalability in resource-constrained settings.

Previous studies [5,16] have largely focused on the analytical components of DSMs such as data structure, modeling assumptions, and quantitative outputs (e.g., ICERs and sensitivity analysis). While such contributions advanced methodological rigor in HTA, they often neglected how decision tools are implemented and institutionalized. By contrast, this review foregrounds procedural dimensions such as transparency, consistency, deliberation, and legitimacy, highlighting the importance of *how* decisions are made, not just *what* is decided. A notable conceptual contribution is the multi-criteria evaluation framework proposed by Kalo et al. [18] for assessing value-added medicines in pricing and reimbursement. While conceptually aligned with procedural priorities—such as unmet medical need, innovation, and adherence—it remains unvalidated in real-world settings. By situating this within a broader evidence base, our review highlights the need not only for robust frameworks but also for empirical testing and contextual adaptation.

This review contributes to the field in several distinct ways: it moves beyond tool identification by evaluating real-world performance and relevance, it introduces a global lens by including studies from underrepresented regions (e.g., Southeast Asia and the Middle East), and it centers governance as intrinsic to pharmaceutical policy effectiveness, reaffirming that how decisions are made is inseparable from what decisions are made, positioning governance and procedural fairness as core components of pharmaceutical policy effectiveness.

Finally, while this review has primarily focused on the existing structured decision tools, it is important to note that future DSMs are expected to evolve rapidly. The incorporation of digital technologies including AI, automated evidence synthesis, and machine learning is already being piloted in some clinical settings. These advancements promise to improve responsiveness and scalability but necessitate careful consideration of transparency, ethical oversight, and governance mechanisms to safeguard accountability and fairness in decision-making processes.

This review has limitations. First, excluding non-English studies may have omitted tools relevant to other linguistic and cultural settings. Second, the review relies on published sources, meaning informal or unpublished decision tools were not captured. Third, due to a lack of consistent outcome metrics, we were unable to perform quantitative comparisons of DSM effectiveness.

Future research should focus on longitudinal studies evaluating how tools like QoDoS affect organizational decision culture over time, on development of hybrid frameworks that combine process and outcome quality measures in pharmaceutical decisions, as well as on standard-setting initiatives across HTA and regulatory bodies to embed core process quality metrics. Collaborative efforts among HTA networks, WHO regional offices, and academic researchers could support methodological harmonization and uptake, especially in resource-constrained settings.

## 5. Conclusions

This narrative review highlights the critical yet often overlooked role of procedural quality in pharmaceutical policy decision-making. While traditional decision support models have focused on clinical and economic outputs, process-oriented tools such as QoDoS, WHO-INTEGRATE, and stakeholder-informed MCDA frameworks emphasize how decisions are made, bringing attention to transparency, stakeholder engagement, contextual relevance, and institutional accountability.

Despite the availability of several validated tools, their adoption and integration remain limited, particularly outside of high-income countries. This review reveals significant variability in methodological rigor, limited empirical validation, and inconsistent reporting of critical factors such as conflict of interest and funding. These findings suggest that process-oriented DSMs are still in a nascent stage of implementation, and their policy impact remains underexplored.

Strengthening the quality of decision-making processes is not merely a technical improvement but a foundational component of sustainable, trustworthy, and equitable pharmaceutical policy. Embedding process-focused DSMs into institutional workflows can enhance legitimacy, consistency, and public trust in pharmaceutical policy, particularly in contexts of uncertainty, limited resources, and rising demand for equitable access to innovation.

## Figures and Tables

**Figure 1 healthcare-13-01861-f001:**
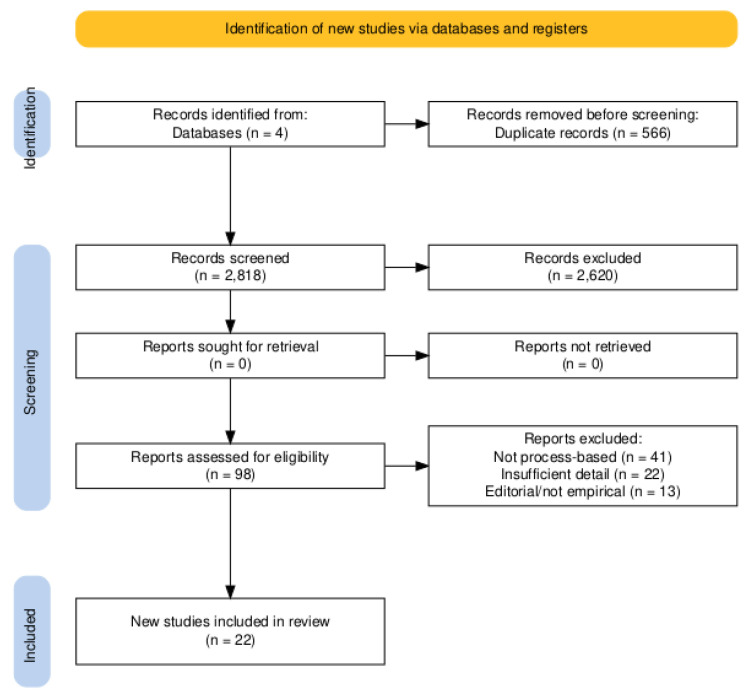
PRISMA Flow diagram: DSMs in pharmaceutical policy [12].

**Table 1 healthcare-13-01861-t001:** Overview of included studies and key characteristics.

Author (Year)	Process-Focused DSM	Country/Context	Policy Domain	Reported Outcomes
Donelan et al., 2021 [9]	QoDoS	UK/Regulatory and HTA	Quality of decision-making	Improved consistency and transparency
WHO, 2019 [10]	WHO-INTEGRATE	Global/LMICs	HTA, UHC policy	Value alignment, contextual integration
Fasseeh et al., 2020 [13]	Stakeholder MCDA	Oman	Pricing, value assessment	Stakeholder alignment, Legitimacy
Bujar et al., 2017 [8]	QoDoS	Global	Industry-regulator interface	Evaluation framework Development
Al- Badriyeh et al., 2016 [15]	QoDoS, MCDA	Kuwait, Quatar	Regulatory process	Quality benchmarking, internal process improvement
Marsh et al., 2014 [16]	EVIDEM	International case comparisons	MCDA in HTA	Structured evaluation criteria, stakeholder integration
Yfantopoulos and Chantzaras, 2018 [17]	Policy analysis	Greece	Pricing reforms	Governance challenges, transparency deficits
Kaló et al., 2021 [18]	Value-Added Meds Eval	EU (pilot cases)	Reimbursement/value	Value optimization and HTA acceptability
Visintin et al., 2019 [19]	HTA framework comparison	7 OECD countries	Reimbursement	Variability in appraisal criteria across HTA bodies
Costa et al., 2022 [20]	Regulatory assessment	EU/US	Access to orphan drugs	Delays in access, regulatory divergence
Laba et al., 2020 [21]	Consumer-engaged MCDA	Canada	Public drug plans	MCDA feasibility, transparency
Moosivand et al., 2021 [22]	AHP-TOPSIS MCDA	Iran	Drug shortages policy	Multi-criteria prioritization framework
McEwin et al., 2025 [23]	Post-market evidence scoping	Australia	Oncology decision-making	Gaps in real-world integration in decisions
Grundy et al., 2022 [24]	Governance and COI analysis	SE Asia Region	Regulatory governance	Conflict of interest management frameworks
Angelis et al., 2018 [25]	HTA practices comparison	8 EU countries	Value assessment	Divergence in criteria and transparency
Pisana et al., 2022 [26]	Real-World Data evaluation	18 European countries	Oncology utilization	RWD potential and limitations in routine decisions
Almomani et al., 2022 [27]	Local HTA capacity	Jordan	Pricing and reimbursement	Institutionalization progress
Hogervorst et al., 2023 [28]	Stakeholder consultation	EU	Regulatory-HTA convergence	Recommendations for Integration
Eskola et al., 2022 [29]	RWE mapping	EMA/EU	Marketing authorization	Frequency of RWD in Approvals
Vitry et al., 2015 [30]	Regulatory ethics review	EU	Adaptive licensing	Evaluation uncertainty, transparency gaps
Sehdev and Chambers, 2022 [31]	CADTH Re-evaluation framework	Canada	Reimbursement revisions	Justification and accountability mechanisms
Abdullah et al., 2019 [14]	Stakeholder MCDA	Kuwait	Procurement	Stakeholder alignment, MCDA

**Table 2 healthcare-13-01861-t002:** Quality appraisal of included studies using process-oriented DSMs).

#	Study/(Author, Year)	DSM Used	Structure	Transparency	Impact	Evaluation	Total/8
1	Donelan et al., 2016 [9]	QoDoS	2	2	2	2	8
2	WHO, 2019 [10]	WHO-INTEGRATE	1	2	2	1	6
3	Fasseeh et al., 2020 [13]	Stakeholder MCDA	1	2	2	0	5
4	Bujar et al., 2017 [8]	Multiple (incl. QoDoS)	2	1	1	1	5
5	Al- Badriyeh et al., 2016 [15]	Adapted QoDoS, MCDA	1	2	1	1	5
6	Marsh et al., 2014 [16]	EVIDEM	1	2	2	1	6
7	Yfantopoulos and Chantzaras, 2018 [17]	Policy Analysis	1	1	1	0	3
8	Kalo et al., 2021 [18]	Value-Added Meds Eval	1	2	1	1	5
9	Visintin et al., 2019 [19]	HTA Framework Analysis	2	2	1	1	6
10	Costa et al., 2022 [20]	Regulatory Pathway Eval	1	1	1	1	6
11	Laba et al., 2020 [21]	MCDA	1	2	2	1	6
12	Moosivand et al., 2021 [22]	MCDM/AHP-TOPSIS	1	1	2	1	5
13	McEwin et al., 2025 [23]	Scoping RWD	1	2	1	1	5
14	Grundy et al., 2022 [24]	Governance/Disclosure	2	2	2	0	6
15	Angelis et al., 2018 [25]	Comparative HTA Review	2	2	1	1	6
16	Pisana et al., 2022 [26]	RWD Evaluation	1	2	1	1	5
17	Almomani et al., 2022 [27]	Local HTA Implementation	2	2	1	1	6
18	Hogervorst et al., 2023 [28]	Qualitative Policy Eval	1	2	2	0	5
19	Eskola et al., 2022 [29]	RWE Mapping	1	2	1	1	5
20	Vitry et al., 2015 [30]	FDA/EMA Decision Review	1	1	1	1	4
21	Sehdev and Chambers, 2022 [31]	CADTH Re-Eval Framework	1	1	1	1	4
22	Abdullah et al., 2019 [14]	Stakeholder MCDA (Kuwait)	2	2	2	1	7

**Table 3 healthcare-13-01861-t003:** Comparative features of core DSMs.

Tool	Transparency	Stakeholder Inclusion	Consistency	Equity Focus	Usability (Applied Use)
QoDoS	High	Moderate	High	Low	High (industry, regulators)
WHO-INTEGRATE	Moderate	High	Moderate	High	Moderate (LMIC policy use)
AGREE II	High	Low	High	Low	High (guideline contexts)
EVIDEM (MCDA)	Moderate	High	High	Moderate	Moderate

## Data Availability

No new data were created or analyzed in this study. Data sharing is not applicable to this article.

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
