# Peer review of "Beyond the Pill: Mapping Process-Oriented Decision Support Models in Pharmaceutical Policy"

_healthcare, 2025, doi:10.3390/healthcare13151861_

Round 1
Reviewer 1 Report
Comments and Suggestions for Authors
The manuscript is systematic review of process-focused decision support models (DSMs) used in pharmaceutical policy. The authors clearly articulate the relevance of procedural quality in decisions surrounding pricing, reimbursement, and access to medicines. The inclusion of both peer-reviewed and gray literature, adherence to PRISMA 2020 guidelines, and categorization of tools based on evaluation dimensions collectively strengthen the methodological rigor of the review. However, the manuscript would benefit from the following clarifications and enhancements:
- While the term “decision support model” is frequently used, a clearer definition or framework distinguishing DSMs from broader decision-making frameworks or checklists would help position the study within the field. Consider including a schematic or typology of DSMs (e.g., evaluative vs. deliberative; qualitative vs. quantitative).
- The narrative synthesis describes tools like QoDoS, WHO-INTEGRATE, and AGREE II, but more detailed comparison—including strengths, limitations, domains assessed, and specific use cases—would enhance the paper’s utility.
- A summary comparison table mapping each tool against dimensions like transparency, stakeholder inclusion, consistency, equity, and usability would be valuable.
- The review notes limited uptake in low- and middle-income countries (LMICs), but further analysis of regional or income-level differences in DSM application could strengthen the policy relevance. Are there contextual barriers (e.g., institutional capacity, resource constraints) affecting DSM implementation in LMICs?
- Please strengthened the discussion how various DSMs have tangibly influenced policy outcomes, implementation timelines, or public trust—where such evidence exists.
- Discuss how digital platforms or AI tools might be integrated into next-generation DSMs.
- Clarify whether studies excluded during full-text review were primarily due to lack of focus on process dimensions or tool evaluation.
- The introduction should supplement the current research and enrich the content of the paper. For example: https://doi.org/10.1186/s13677-024-00675-z,
https://doi.org/10.1016/j.hrtlng.2025.03.008, https://doi.org/10.34172/ijhpm.2024.8150
Author Response
Reviewer 1
Comments 1: While the term “decision support model” is frequently used, a clearer definition or framework distinguishing DSMs from broader decision-making frameworks or checklists would help position the study within the field. Consider including a schematic or typology of DSMs (e.g., evaluative vs. deliberative; qualitative vs. quantitative).
Response 1: We appreciate the reviewer's valuable suggestion. In response, we have clarified our definition of decision support models (DSMs) and explicitly distinguished them from broader decision-making frameworks in the Introduction section. We now define DSMs as structured, evidence-informed approaches used to support healthcare policy decisions, and we introduce a two-dimensional typology (evaluative vs. deliberative; quantitative vs. qualitative) to better position DSMs within the field (see lines 49-55 in the revised manuscript).
Comments 2: The narrative synthesis describes tools like QoDoS, WHO-INTEGRATE, and AGREE II, but a more detailed comparison including strengths, limitations, domains assessed, and specific use cases would enhance the paper’s utility.
Response 2: We appreciate the reviewer's thoughtful feedback. We have revised the Discussion section (see lines 257-265) to incorporate a comparative analysis of the three most frequently cited tools: QoDoS, WHO-INTEGRATE, and AGREE II in terms of how each contributes a distinct lens to process quality evaluation.
Comments 3: A summary comparison table mapping each tool against dimensions like transparency, stakeholder inclusion, consistency, equity, and usability would be valuable.
Response 3: We sincerely appreciate the reviewer's helpful recommendation to include a conceptual summary table comparing QoDoS, WHO-INTEGRATE, and AGREE II across five procedural dimensions. Table 3 has been included in the Discussion section as a reflective synthesis of our findings (see page 8). We have clarified that it does not represent a formal scoring exercise but aims to illustrate key differences in tool orientation and applicability, supporting a more nuanced understanding of their comparative utility (see lines 266-272).
Comments 4: The review notes limited uptake in low- and middle-income countries (LMICs), but further analysis of regional or income-level differences in DSM application could strengthen the policy relevance. Are there contextual barriers (e.g., institutional capacity, resource constraints) affecting DSM implementation in LMICs?
Response 4: We appreciate the reviewer for raising this important point. We have now expanded the Discussion section (see lines 313-321) to explicitly address contextual barriers affecting DSM implementation in LMICs, such as institutional capacity and resource constraints. We also highlight relevant case studies and propose that future DSM development should prioritize contextual feasibility and scalability.
Comments 5: Please strengthened the discussion how various DSMs have tangibly influenced policy outcomes, implementation timelines, or public trust—where such evidence exists.
Response 5: We appreciate the reviewer’s point and have added a paragraph in the Discussion section noting existing but limited evidence on DSM impact, e.g., QoDoS use in regulatory agencies and MCDA in public reimbursement decisions. We also emphasize the absence of long-term evaluation data and call for embedded feedback mechanisms (see 306-312).
Comments 6: Discuss how digital platforms or AI tools might be integrated into next-generation DSMs.
Response 6: We appreciate the reviewer's important comment regarding the future evolution of DSMs and the integration of digital technologies such as AI and automated evidence synthesis. We have incorporated this perspective into the Discussion section, highlighting early developments in AI-driven tools and the emerging challenges of transparency, explainability, and governance. The paragraph emphasizes the importance of embedding such tools within robust accountability frameworks to ensure trust and procedural fairness in future DSM use (see lines 341-347).
Comments 7: Clarify whether studies excluded during full-text review were primarily due to a lack of focus on process dimensions or tool evaluation.
Response 7: We appreciate the reviewer's important observation. In response, we have revised the Methods section accordingly (see lines 175-181, at the end of the Study Selection section).
Comments 8: The introduction should supplement the current research and enrich the content of the paper. For example: https://doi.org/10.1186/s13677-024-00675-z
Response 8: We thank the reviewer for this important observation to enrich the introduction and connect it more explicitly to recent research developments. We have revised the Introduction section, accordingly, integrating relevant insights from the literature (see lines 78-86).
Reviewer 2 Report
Comments and Suggestions for Authors
This narrative review examines process-focused decision support models (DSMs) applied in pharmaceutical policy, particularly in pricing, reimbursement, and access to medicines. Drawing on studies published between 2000 and 2025, the review identifies key tools such as QoDoS, WHO-INTEGRATE, and AGREE II, highlighting their contributions to improving transparency, stakeholder engagement, and procedural legitimacy. While QoDoS stands out for its broad application and validation, many models lack consistent implementation and empirical evaluation, especially in low- and middle-income countries. The review underscores the importance of process quality in decision-making and calls for further standardization and broader adoption of DSMs to enhance trust and sustainability in pharmaceutical governance.
The article would benefit from presenting the complete search strings used for each database, as well as providing a detailed flowchart or table indicating the number of studies at each screening phase, including the number of exclusions and reasons for exclusion. Including this level of detail would greatly enhance the reproducibility and transparency of the review.
Additionally, i recommend that the authors consider elaborating a structured abstract, clearly delineating sections such as Background, Objectives, Methods, Results, and Conclusions. This would improve clarity and help readers quickly grasp the study’s scope and key findings.
Author Response
Reviewer 2 |
Comments 1: The article would benefit from presenting the complete search strings used for each database, as well as providing a detailed flowchart or table indicating the number of studies at each screening phase, including the number of exclusions and reasons for exclusion. Including this level of detail would greatly enhance the reproducibility and transparency of the review. |
Response 1: We appreciate the reviewer for bringing this to our attention. Therefore, we have updated our manuscript in the Methods section (see lines 112-118) and in the Results section (see lines 175-181 and page 4 including Fig 1). |
Comments 2: Additionally, I recommend that the authors consider elaborating a structured abstract, clearly delineating sections such as Background, Objectives, Methods, Results, and Conclusions. This would improve clarity and help readers quickly grasp the study’s scope and key findings. |
Response 2: We appreciate the reviewer's valuable comment. We have, accordingly, modified the Abstract section. |